# Cannabinoid Therapeutic Effects in Inflammatory Bowel Diseases: A Systematic Review and Meta-Analysis of Randomized Controlled Trials

**DOI:** 10.3390/biomedicines10102439

**Published:** 2022-09-29

**Authors:** Antonio Vinci, Fabio Ingravalle, Dorian Bardhi, Nicola Cesaro, Sara Frassino, Francesca Licata, Marco Valvano

**Affiliations:** 1Hospital Health Management Area, Local Health Authority “Roma 1”, 00133 Roma, Italy; 2Hospital Health Management Area, Local Health Authority “Roma 6”, 00041 Albano Laziale, Italy; 3Post-Graduate School of Hygiene and Preventive Medicine, University of L’Aquila, 67100 L’Aquila, Italy; 4Gastroenterology, Hepatology and Nutrition Division, Department of Life, Health and Environmental Sciences, University of L’Aquila, 67100 L’Aquila, Italy; 5Department of Health Sciences, School of Medicine, University of Catanzaro “Magna Græcia”, 88100 Catanzaro, CZ, Italy

**Keywords:** inflammatory bowel disease, cannabinoid, supplementation therapy

## Abstract

(1) Introduction: Inflammatory Bowel Disease (IBD) patients may benefit from cannabinoid administration supplementary therapy; currently no consensus on its effect has been reached. (2) Methods: a systematic review of RCTs on cannabinoid supplementation therapy in IBD has been conducted; data sources were MEDLINE, Scopus, ClinicalTrials. (3) Results: out of 974 papers found with electronic search, six studies have been included into the systematic review, and five of them, for a grand total of 208 patients, were included into the meta-analysis. (4) Conclusions: cannabinoid supplementation as adjuvant therapy may increase the chances of success for standard therapy of Crohn’s Disease during the induction period; no statement on its potential usage during maintenance period can be derived from retrieved evidence. Its usage in Ulcerative Colitis is not to be recommended. If ever, low-dose treatment may be more effective than higher dosage. Mean CDAI reduction was found stronger in patients treated with cannabinoids (mean CDAI reduction = 36.63, CI 95% 12.27–61.19) than placebo. In future studies, it is advisable to include disease activity levels, as well as patient-level information such as genetic and behavioral patterns.

## 1. Introduction

### 1.1. Background

Inflammatory bowel disease (IBD), including both ulcerative colitis (UC) and Crohn’s Disease (CD) are chronic, idiopathic inflammatory diseases, causing inflammation of the gastro-intestinal tract [1,2]. Typically, the disease course is characterized by the alternation of periods of remission and flare-ups [3].

There are several scoring systems presently available to classify disease severity in both CD and UC. Crohn’s Disease Activity Index (CDAI) and Mayo score are among the most used in both clinical practice and randomized clinical trials (RCT) [4]. However, it is important to underline that evaluation of CD or UC is based on a combination of clinical, biochemical, stool, endoscopic, cross-sectional imaging, and histological investigations [5]. In fact, according to the current treat-to-target therapeutic strategy, there is a multitude of potential therapeutic targets (such as a clinical, endoscopic, and histological activity) and achieving a multi-parametric remission allows achieving a long term deep remission and a better quality of life [6,7,8]. Pain reduction play also an important role in IBD management, as it is nowadays common knowledge that inappropriate management of pain creates very important physical, psychological and social consequences, especially in the most debilitating forms of pain—such as chronic ones [9].

IBD has become a global disease with accelerating incidence in the last decades; therefore, there is a need to improve the innovations in the delivery of care [10]. In addition to conventional therapy (including mesalamine, and an immuno-modulator, such as azathioprine or methotrexate), new therapeutic options, including biological therapy and small molecules, represent a promising strategy for IBD treatment. In fact, after the introduction of the biological therapy, a growing number of new molecules have become available [11]. However, the ultimate goal to achieve a stable clinical remission is far from being reached. In this scenario, a growing number of adjunctive therapies, such as vitamin D and probiotics, has been proposed over the years, integrating the current standard of care in the hope of achieving clinical benefits [12,13,14].

Another potential treatment as adjuvant therapy is represented by cannabinoid administration [15]. Available data have shown a possible role of cannabis as a supportive medication, particularly in pain reduction; however, it remains unclear whether cannabinoids have any impact on the underlying inflammatory process of IBD, despite the evidence of their anti-inflammatory effect in vitro and in animal models of IBD [16,17,18]. Some evidence has been presented, that it may mitigate several of the well-described complications of CD among hospital inpatients [19]. In the U.S., a nationwide analysis did not show any benefit in terms of hospital readmission for IBD-specific causes, although cannabis use was associated with reduced 30-day hospital readmission rates for all causes [20].

The anti-inflammatory role of the Endo-Cannabinoid System (ECS) is already well known. It is implicated in gut homeostasis, modulating gastrointestinal motility, visceral sensation, and inflammation, as well as IBD pathogenesis. ECS molecular targets include, in addition to the cannabinoid receptors, transient receptor potential vanilloid 1 receptors, peroxisome proliferator-activated receptor alpha receptors and the orphan G-protein coupled receptors, GPR55 and GPR119 [21]. Additionally, epigenetic modification has been demonstrated to be induced by cannabinoids, and to mediate inflammation processes [22]. Some pharmaceutical agents have already been investigated, and have shown some potential in pre-clinical studies [23]. In particular, Cannabidiol has been proven effective as a therapeutic option for oral wound healing in vivo mouse models, both in Chemotherapy-Induced Oral Mucositis (via the Nrf2/Keap1/ARE Signaling Pathways) and on wound-induced ulcers [24,25]. Some patients also use cannabinoids in order to alleviate their symptoms, and while effective, a higher usage has been associated with a higher risk of surgery in patients with Crohn’s disease [26].

Two Cochrane Systematic reviews published four years ago attempted to assess the efficacy and safety of cannabis and cannabinoids for the treatment of patients with UC and CD, respectively, but did not perform pooling of results and came to no definitive conclusion on their role as adjuvant therapy [27,28].

### 1.2. Study Rationale and Hypothesis

A recent meta-analysis on the matter conducted by Doeve BH et al. investigated the role of cannabinoid supplementation in inducing biomarker variations, improving quality of life, and changing in risk for selected hospital outcomes [29]. However, as the authors themselves acknowledged, it was not exempt from limitations, most notably the high heterogeneity in the design of included studies (both observational and interventional studies were included in the analysis). Moreover, there was no clear-cut clinical outcome definition (in terms of chances of success of adjuvant therapy administration), nor did this work have the objective to quantify the potential benefit of such therapy integration. In addition, in the last two years some new evidence has been added to literature. Likewise, there is no evidence on recommended dosages of administration ways, mostly due to the scarcity of available literature on the matter.

For these reasons, it was chosen to perform a meta-analysis focusing on binary outcomes of therapy supplementation, and to try to quantify the presumptive benefits of cannabinoids administration in reducing disease severity.

The Null Hypothesis (H_0_) is that there is no impact of cannabinoids supplementation in clinical outcome, nor in disease activity reduction.

### 1.3. Objectives

The primary objective of the present study is to understand if cannabinoid supplementation to standard therapy is favorable, in terms of clinical outcome, in IBD patients, investigating if any difference is found between CD and UC patients.

The secondary objective is to quantify the effects of cannabinoid administration in reducing objective disease severity, as defined by clinical standard practice scores.

## 2. Methods

This systematic review and meta-analysis was performed following a protocol designed a priori, registered on the Open Science Framework (OSF) Registries, https://doi.org/10.17605/osf.io/q9mwz (accessed on 10 August 2022).

The research question has been developed using PICO; “P” stands for Patients, “I” as Intervention, “C” as Comparison and “O” as Outcome. PICO items were defined as per Table 1.

### 2.1. Stakeholder Involvement (PPI)

No patient was involved in the conduction of present study. No patient and public involvement has been planned or sought in the conduction of this systematic review.

### 2.2. Eligibility Criteria

Studies meeting the following criteria were included into systematic review:

Study design:○Randomized Controlled Trial (RCT);○Prospective case-control study;

Exposure:○Treatment with cannabinoids, by either inhalation or oral administration.

Outcome:○Clinical evaluation of the patient, either via standard-use scale or via other validated clinical measurement.

Publication type:○Primary studies published in peer-reviewed journals;○Non-peer reviewed publications and grey literature articles (e.g., trial results), as long as they are publicly available.

Studies meeting the following criteria were excluded from the systematic review:

Study design:○Cross-sectional (prevalence) studies;○Case reports and case series;○Prospective and retrospective studies without measures of association and confidence intervals (or data enabling to calculate them);○Studies with no comparison.

Publication type:○Abstracts with no data available (i.e., conference abstracts, unless providing data for analysis);○Studies reporting results that have been superseded by subsequent reports from the same study (note: this includes conditions when that reports include a mix of results that have or have not been updated).

Meta-analysis would include only studies whose data (either reported or retrieved after request to the respective corresponding author) allowed the authors to perform required calculations.

### 2.3. Information Sources and Search Strategies

MEDLINE, Scopus and ClinicalTrials.gov (accessed on 10 August 2022) databases were searched electronically on 15 May 2022, using combinations of the relevant medical subject heading terms, key words and word variants, as shown in Table 2, as well as the filters used for search optimization. The choice of which database to search was driven by their availability to the authors.

### 2.4. Addressing Bias

Mathur and Van der Weele recommendations were followed in order to address bias in this work, although most of them were not implemented since they were aimed at meta-analyses of Non-Randomized Studies (NRS) [30]. Specifically:It was chosen to include only prospective controlled studies (both RCT and NRS) in the selection process, since they provide the least biased results while still permitting reasonable statistical precision;Risk of Bias was rated using a specific, validated tool in our protocol;Sensitivity analyses were performed by executing a leave-one-out analysis, consisting in performing multiple meta-analyses by excluding one study at each step, and a meta-regression [31].

### 2.5. Selection Process

Two different authors (AV and FI) independently screened the article titles and abstracts in each database. Duplicates were removed in this phase.

Disagreements on inclusion/exclusion were discussed by the authors and resolved by consensus or by recourse to a third author (DB). Studies were then labeled for inclusion or exclusion.

Every article meeting the eligibility criteria was considered for subsequent qualitative synthesis. Two different authors (DB, FI) independently screened article content and studies were henceforth labeled for inclusion or exclusion in the systematic review. Disagreements were discussed by the authors and resolved by consensus or by recourse to a third author (MV). The selection process described above has been summarized in a flow diagram, shown in Figure 1.

### 2.6. Data Collection Process

Two different authors (AV and FI) independently collected data from retrieved articles. Studies with missing or unclear data were included in the narrative synthesis but excluded from meta-analysis. Disagreements were discussed by the authors and resolved by consensus or by recourse to a third author (FL).

### 2.7. Data Items

For every study included in the meta-analysis, the following data were retrieved:Publication year;Involved country;Disease type;Treatment type and duration;Control type;Follow-up time;Study effect measure;Number of patients in each cohort;Therapeutic successes in each cohort;Therapeutic failures in each cohort.When available, variation in disease activity index in each cohort was also retrieved.

Therapeutic success was defined as either disease relapse, supported by objectifiable findings (such as absence of blood in stool), or by downgrading in disease severity, as measured by any standard-practice scale in use. Any other outcome was considered as therapeutic failure, including reduction in any disease activity index that did not translate in downgrading to a lower class (meaningless reduction). Losses to follow-up were considered as therapeutic failure, as per Intention-To-Treat (ITT) analysis.

In case of missing or unavailable data, a message with data request was sent to the corresponding author. This occurrence happened only once, and the authors promptly answered to our request.

### 2.8. Study Risk of Bias Assessment

The revised Cochrane Risk of Bias 2.0 (RoB 2.0) tool was used for assessment of Risk of Bias. Two authors (FI and DB) independently applied the tool on the retrieved studies. Disagreements were discussed by the authors and resolved by consensus or by recourse to a third author (FL).

### 2.9. Effect Measures

The effect measure used for quantitative synthesis of primary outcome was pooled Risk Ratio (RR). Mean Difference (MD) in CDAI reduction was the effect measure used for secondary outcome. Other information incidentally retrieved from included articles (such as, secondary endpoint results or bioptic evidence) were not used for meta-analysis but were considered in narrative synthesis.

### 2.10. Synthesis Methods

All studies responding to inclusion criteria were selected for narrative synthesis.

Results have been reported both narratively and graphically, and have been summarized in tables when deemed appropriate.

Given heterogeneity in retrieved studies design and the fact that each study was performed on different populations, the authors decided to perform a meta-analysis pooling of risk ratios and mean differences using the Random-effects Restricted Maximum Likelihood (REML) model. In order to quantify statistical heterogeneity, the *I*^2^ statistic was nonetheless calculated [32].

In order to minimize Type I error risk, Intention to Treat (ITT) analysis was performed, considering all dropout individuals as unsuccessful treatment.

The equations provided by the Cochrane Handbook for Systematic Reviews of Interventions, (Version 6.3, 2022) were used to calculate mean differences in CDAI reduction [33].

Subgroup pooling depending on disease type (CD or UC) was performed.

Leave-one-out analysis was also performed, in order to investigate the influence of each study on the overall effect-size estimate and to identify influential studies.

Meta-regression was performed in order to explore potential causes of heterogeneity and investigate moderator variables; in particular, analysis was performed between primary outcome effect and supplementation dosage. Dosage was calculated as daily CHB administration (in mg), regardless of the medium, and maximal dosage were considered in case of scaled-up therapy.

Normally, the administration route plays a great role in any drug distribution. THC and cannabidiol pharmacokinetics are quite complex to predict: smoked cannabis absorption has a quick concentration peak but is heavily influenced intra- and inter-subject variability in smoking dynamics, which contributes to uncertainty in dose delivery. The number, duration, and spacing of puffs, hold time, and inhalation volume, greatly influences the degree of drug exposure. Oral administration is influenced by several factors, including variable absorption, degradation of drug in the stomach, and significant first-pass metabolism to active 11-OH-THC and inactive metabolites in the liver [34]. Nonetheless, smoked peak concentration reaches quickly a low plasmatic availability (within 1 h), and similar concentrations were reported for standard oral intake of synthetic THC (10 mg THC) [35]. Concerning the long-term effects of cannabinoid as an anti-inflammatory adjuvant in the gastrointestinal tract, we believe that difference in administration route does not hinder the validity of meta-regression analysis, as differences in short-term peak concentration values should only matter for the effects on the nervous system [36].

Stata^®^ SE v.17.0 and Microsoft Excel MSO 2016 were used for calculations and graph drawing.

### 2.11. Reporting Bias Assessment

In order to assess the risk of bias derived from missing publications in the synthesis (publication bias), a graphical assessment via funnel plot has been produced, along with a L’Abbé plot [37].

### 2.12. Certainty Assessment

The Grading of Recommendations Assessment, Development and Evaluation (GRADE) approach has been used in order to assess the certainty of the evidence [38].

Risk of Bias has been evaluated using the RoB 2.0 tool.

Inconsistency has been evaluated by calculating the *I*^2^ statistic, as described by Higgins and Thompson [39]. *I*^2^ quantifies the proportion of variation in the point estimates due to differences between studies [40].

Indirectness was excluded by protocol, by selecting only studies with well-defined population, outcomes and treatment protocol.

Confidence Intervals (Cis), *I*^2^ and *p*-values were calculated in order to highlight potential precision issues. The Harbord test has been used in order to investigate small studies effects in meta-analysis. This regression-based test is used for binary data with effect sizes log odds-ratio and log risk-ratio [41].

### 2.13. Reporting

This systematic review and meta-analysis was reported according to the 2020 Preferred Reporting Items for Systematic Reviews and Meta-Analyses (PRISMA) guidelines [42]. Its checklist is available as Appendix A.

## 3. Results

### 3.1. Study Selection

Results of searches and selection process are shown in Figure 1. Overall, 974 studies were retrieved from databases, and after screening sessions 26 studies were initially retrieved for full evaluation. Twenty-one studies were excluded for several reasons (see Figure 1 for details) and 1 study was retrieved from other studies’ bibliography.

At the end, six studies were included in the systematic review. All of them were deemed fit for inclusion in meta-analysis.

### 3.2. Study Characteristics

Five retrieved studies were selected for synthesis in meta-analysis. One further study was selected from references of retrieved records. All retrieved studies were RCTs.

A grand total of 227 people was included in the synthesis, with studies coming from two countries (UK and Israel). UC patients from Matalon 2021 were not included in meta-analysis due to data unavailability for this cohort: 208 people were included in the meta-analysis.

All studies were carried on for several weeks, with a follow-up period ranging from 8 to 10 weeks. According to GRADE recommendations, characteristics of included studies and summary of findings are represented in Table 3.

### 3.3. Risk of Bias in Studies

The RoB 2.0 tool was used for the evaluation of the risk of bias and confounding [49]. Overall, most studies were of average quality, with two studies (33%) being at Low Risk, three studies (50%) at Some Concern, and only one (16%) being at High Risk of bias.

RoB results are shown in Table 4 and graphically summarized in Figure 2.

### 3.4. Results of Individual Studies

Results of retrieved studies are summarized, alongside their characteristics, in Table 3, and have been reported narratively below. All Confidence Interval (CI) values reported in brackets are calculated as 95% CI.

Effectiveness of cannabinoid supplementation on clinical outcome

All included studies allowed us to define clinical outcome, either as complete remission or class disease downgrade. Overall, the effect of supplementation therapy on clinical outcome does not seem at all strong, and it is unclear if cannabinoids can play a meaningful role in defining disease outcome. CD patients appear to be the most likely among those who may receive a beneficial effect, although no single study can claim to reach any statistically significant conclusion.

Effectiveness of cannabinoid supplementation on disease activity reduction, quality of life and other variables

Three studies investigated variables different from clinical outcome or disease activity index. Two of them investigated quality of life (QOL) and both concluded that it improved in patients who were in the intervention arm. One study investigated blood samples at study end, along with bioptic samples, and concluded that endocannabinoid presence was higher in the intervention arm, while inflammation markers were higher in the control arm. Overall, evidence seems to favor treatment in regards of outcomes other than primary endpoint.

### 3.5. Results of Synthesis

Meta-analysis results have been summarized with two forest plots. The results show that cannabinoid supplementation, compared to placebo, had a small chance of delivering a therapeutic success (pooled log RR = 0.38, CI 95% −0.004–0.76), if any. This effect was more evident in CD than UC (Figure 3). Moreover, when analyzing CDAI reduction in studies about CD (Figure 4), it was found that mean CDAI reduction was stronger in patients treated with cannabinoids (mean CDAI reduction = 36.63, CI 95% 12.27–61.19).

### 3.6. Risk of Reporting Bias in Syntheses

Publication bias for studies included in the meta-analysis has been evaluated using a purposed funnel plot (Figure 5) that evidenced presence of publication bias for low-power studies going against intervention.

The regression-based Harbord test for small-study effects yields a *p*-value of 0.81, which means that, despite evidence of potential publication bias, the null hypothesis of no small-study effects cannot be rejected.

### 3.7. Certainty of Evidence

The authors believe that usage of a random effects model is the most appropriate for this type of study, its result conceptually being the most likely value in a distribution of results, rather than focusing on finding of a “true” value (as it is the case in unweighted analysis, or in FE models). This choice is optimal because the single included studies are not drawn from the same population, and do not follow the exact same protocol, despite being similar enough for being included in the statistical analysis [50,51].

The I^2^ test shows that low heterogeneity (<40%) was present across included studies.

Sensitivity analysis carried on with the leave-one-out method, showed that exclusion of any particular study, while influential for statistical significance of the overall model, does not alter the results in a significant way (Figure 6).

Each study effect size was also graphically shown via a L’Abbé plot (Figure 7), showing the collocation of each study in regard of intervention and control arms results.

Meta-regression analysis between dosage and retrieved RR was performed. A log regression model was proposed, and it performed discretely in data fitting (Figure 8), suggesting that a better response is obtained on relatively low CHB dosage (R^2^ = 0.16).

## 4. Discussion

### 4.1. Results in Context

Evidence on cannabinoid effectiveness in primary outcome (disease remission or major reduction in symptoms) in IBD is mixed at best. While we observed, in included RCTs, that UC patients did not benefit from cannabinoid therapy in any significant way compared to placebo controls, there is doubt on a potential benefit of cannabinoid supplementation in CD patients. Pooled results were close to statistical significance, which is remarkable considering the relatively low sample size of CD patients in pooled results. Moreover, CDAI reduction was found to be statistically significant in this group of patients. Heterogeneity among studies was low, and this was expected since most studies had very similar design, and were conducted on presumably similar populations.

Sensitivity analysis on primary outcome shows that the main result is robust enough to be found, even if one random study is ignored; however, there is a different impact on statistical significance of the result.

### 4.2. Limitations of Included Studies

This is the most updated meta-analysis on the matter, including all published data of RCTs on cannabinoids usage. It took data only from RCT, in order to ensure the highest quality level possible, but its conclusions are limited by the scarcity of trials on the matter, especially coming from different population settings: most of the primary evidence has actually been produced, over the years, by a single research group. Globally, we believe that included studies had some limitations, mostly regarding clear identification and exclusion of potential confounding factors. Specifically, we noted that there is little information on previous smoking habits, and there is little mention in most studies of baseline therapy. Even when it is mentioned, often there are differences in the therapeutic baseline regimen given to patients within the same study, since they may be enrolled on different disease stages, gravity of illness, and with a different grade of steroid resistance. This made results adjustment impossible in both primary and secondary analyses.

Another important limitation of included studies derives from their low sample size. Even though most included studies had a good randomization process, with an assessed low risk of bias in this domain, their small sample size does not guarantee a successful randomization when taking into account for all possible confounding factors. A large RCT would likely overcome this limitation.

### 4.3. Limitations of the Review Methods

The main limitation of this review lies in the lack of statistical adjustment for potential confounding factors. No such adjustment could be performed, due to the lack of adequate detail in data collected from primary studies. Most studies actually reported data in detail only on primary and secondary outcomes, and reporting of aggregate data on population characteristics made impossible any complex modelization. Nonetheless, in order to investigate for the role of possible moderator or confounding variables, a meta-regression analysis was performed on RR and dosage of CHB. Given the different regimen adopted by the included RCTs, we chose to define daily regimen as the quantitative of administered CHB (in milligrams), regardless of the medium, and to consider maximal dosage in cases when therapy was scaled up to a threshold. The difference is particularly evident when considering that there is a 20-fold difference between the lowest administered daily dosage (Naftali 2013 and Matalon 2021, both 23 mg/day) and the highest (Irving 2018, 500 mg/day).

A log meta-regression model hints that low dosage of CHB may yield better outcomes than massive dosage (R^2^ ≈ 0.16). This is coherent with current literature findings, that suggest the possibility of a non-linear response to CHB [52].

### 4.4. Implications

To our knowledge, this is the only secondary study that analyzes cannabinoid effectiveness as supplementation therapy in IBD and gives an estimation of CDAI reduction. The only other meta-analysis on the topic was performed by Doeve et al., [24] who included in their work both experimental and observational trials, but could not give an estimation of disease activity index variation. The present one is the first work that gives a separate estimation of overall RR of successful cannabinoid supplementation in both CD and UC. In addition, in CD patients, we provided an estimation of CDAI expected reduction after cannabinoids supplementation (36.63 points, CI 95% 12.27–61.19). This data can be used for power size estimation in future RCTs. However, CDAI reduction should be only accounted for in investigation of secondary outcomes due to the very nature of the CDAI index, which can yield a high value even if pain is not the main symptom.

We believe that cannabinoid supplementation may have a role in management of IBD patients, especially concerning pain control; however, there is no evidence to support the hypothesis that it may have a direct anti-inflammatory role in UC patients. Cannabinoids may be useful as substitute of other pain-controlling drugs; however, no study has been performed evaluating potential side effects, especially regarding psychological effects or addiction. Most importantly, there is no RCT up to date testing cannabinoid supplementation vs. standard painkiller. Given that cannabinoid usage is positively correlated with inpatient opioid dose exposure in hospitalized patients with IBD, we feel this is a point worth being investigated in future trials [53]. Likewise, genetic information regarding disease patterns, as well as CHB metabolism and likelihood of side-effects development, could be an added asset in developing targeted therapies [54]. As experience with coronary heart disease demonstrates, such assessment can be improved by combining family history with behavioral and clinical risk factors [55]. This is likely to be an equally valuable strategy if polygenic analyses are involved [56,57].

Given the current results, we recommend that future studies may clearly separate clinical disease activity from pain reduction effects in their outcome assessment, in order not to let any form of bias arise from spurious association. Further patent-level element, such as genetic and behavioral patterns, may also be included as potential confounders or predictors of therapy response.

In regards of clinical practice, cannabinoid therapy should not be considered as the standard analgesic option. Its use should only be restricted to enhancing the results of conventional treatment in causing clinical and laboratory remission during induction period. No statement on its potential usage during maintenance period can be derived from retrieved evidence.

Cannabinoid supplementation as adjuvant therapy has the potential to increase the chances of success for standard therapy of Crohn’s disease, although evidence is still weak. On the other hand, its usage in ulcerative colitis is not to be recommended.

Further studies should focus on confirming this potential and on defining optimal dosages and posology; current literature findings support the option of low dosages, between 10 and 30 mg/day, subdividing the dosage in order to reach an optimal plasma availability (i.e., twice daily). It is not possible to estimate the optimal treatment duration from current literature since all RCTs had a very similar duration; henceforth, should the treating physician elect to initiate cannabinoid therapy, patients should be re-evaluated after 8 weeks for evidence of response to treatment.

Given current pharmacological knowledge, it is to be expected that oral cannabidiol administration will be the most optimal administration route, since it is less reliant on smoking habits and behavior, and since it does not have psychoactive effects, unlike THC [58]. Smoking itself is known to be a major risk factor for flare-ups in CD, and as such potential smoked cannabis benefits should be carefully weighted, especially given the alternate route availability [59]. Until more precise insights are available from primary studies, further consideration should also be made, on a case-by-case basis, depending on the presence of adherences and general gut motility, since they are also influenced by the endocannabinoids system [60].

## Figures and Tables

**Figure 1 biomedicines-10-02439-f001:**
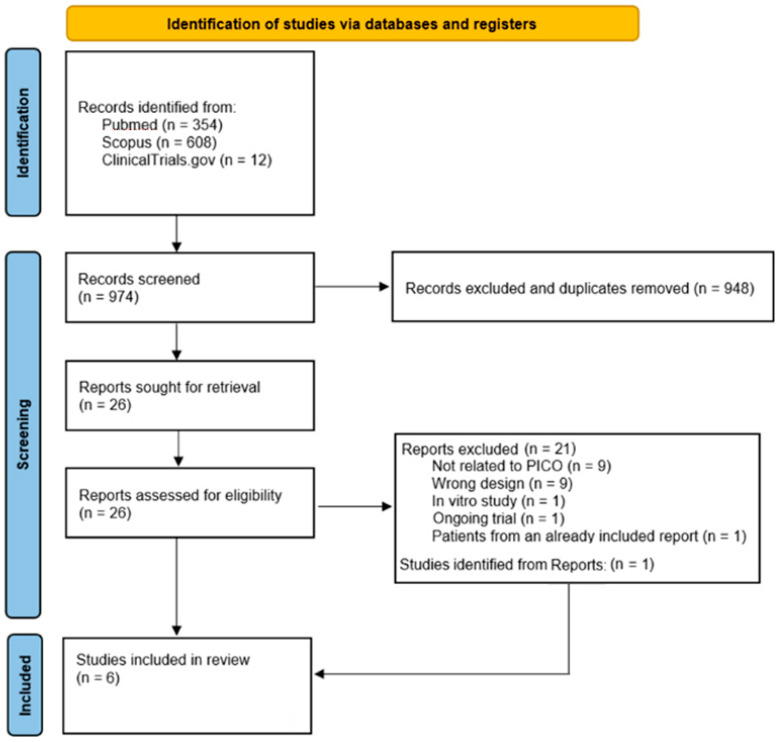
Flowchart of study selection process.

**Figure 2 biomedicines-10-02439-f002:**
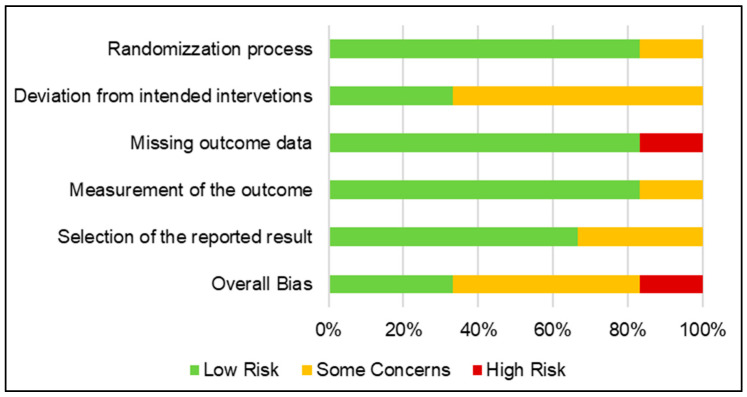
Quality assessment of retrieved studies.

**Figure 3 biomedicines-10-02439-f003:**
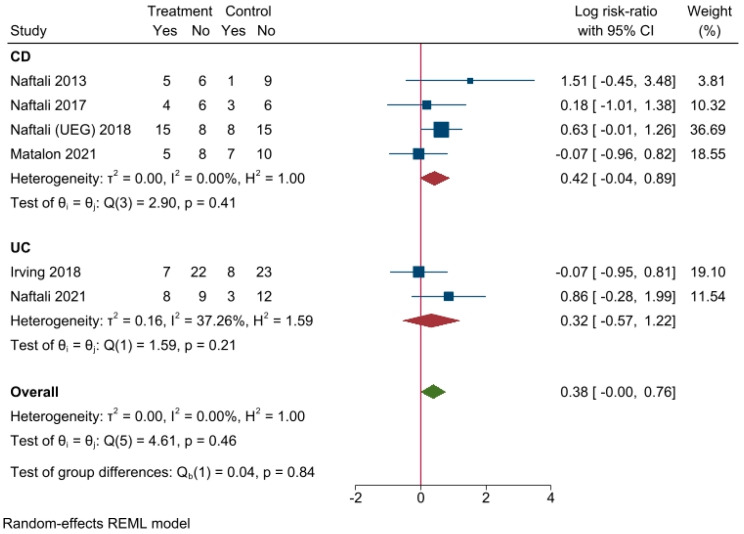
Forest plot of included studies. Values > 0 favors intervention, values < 0 favors control. CD: Crohn’s disease. UC: ulcerative colitis. ITT: intention-to-treat [43,44,45,46,47,48].

**Figure 4 biomedicines-10-02439-f004:**
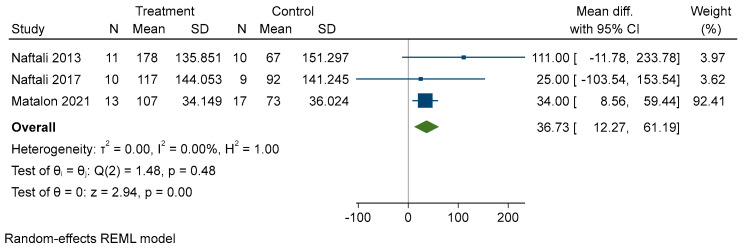
Forest plot of CDAI reduction. Values > 0 favors intervention, values < 0 favors control. SD: standard deviation [43,44,45].

**Figure 5 biomedicines-10-02439-f005:**
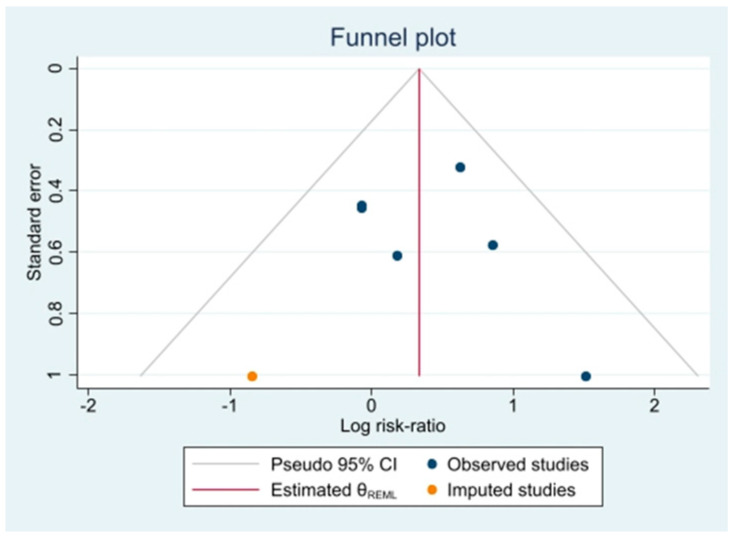
Funnel plot of publication bias assessment.

**Figure 6 biomedicines-10-02439-f006:**
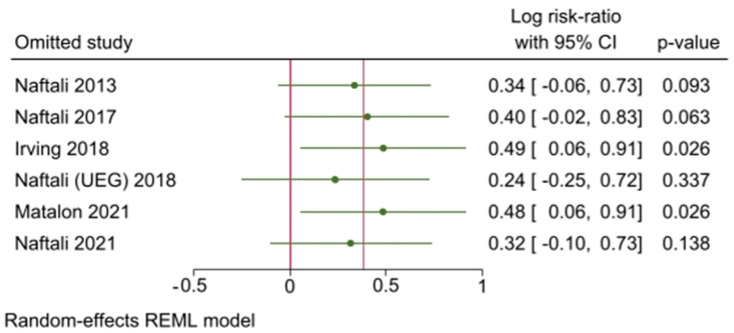
Leave-one-out analysis plot. Values > 0 favor treatment, values < 0 favor control [43,44,45,46,47,48].

**Figure 7 biomedicines-10-02439-f007:**
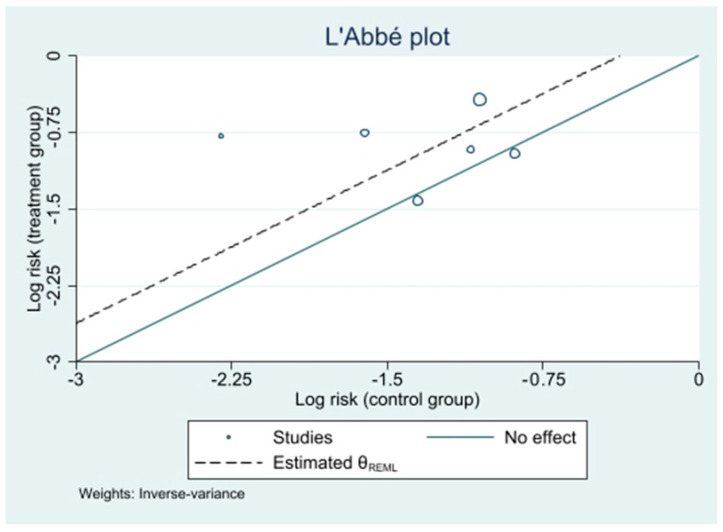
L’Abbé plot.

**Figure 8 biomedicines-10-02439-f008:**
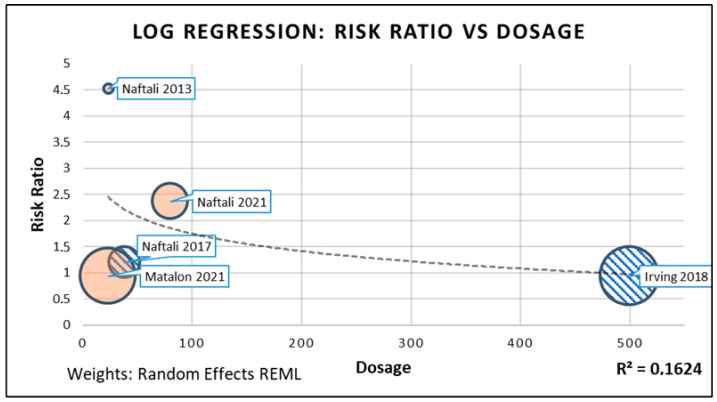
Log meta-regression. Red bubbles represent smoked administration of cannabinoids; blue-barred bubbles represent oral administration of cannabinoids.

**Table 1 biomedicines-10-02439-t001:** PICO query items.

**Population**	Patients with IBD
**Intervention**	Cannabinoid administration
**Comparison**	Standard treatment
**Outcome**	Improvement in patient’s clinical condition

**Table 2 biomedicines-10-02439-t002:** Reference lists of relevant articles and reviews were hand-searched for additional reports.

Database	Search String(s)	Filters
MEDLINE	((ulcerative colitis) OR (crohn) OR (inflammatory bowel disease) OR colitis) and (Cannabis or cannabinoids or THC or cannabidiol)	Publication year: >2000
Scopus	((ulcerative AND colitis) OR (crohn) OR (inflammatory AND bowel AND disease) OR colitis) AND (cannabis OR cannabinoids OR thc OR cannabidiol) PUBYEAR > 2002 AND PUBYEAR > 2001 AND (LIMIT-TO (DOCTYPE, “ar”)) AND (LIMIT-TO (EXACTKEYWORD, “Controlled Study”) OR EXCLUDE (EXACTKEYWORD, “Animals”) OR EXCLUDE (EXACTKEYWORD, “Animal Experiment”))	Not Applicable (already included in string search)
ClinicalTrials.gov (accessed on 10 August 2022)	Disease Type: Inflammatory Bowel Disease (automatic field) Additional keywords: Cannabis	Not Applicable

**Table 3 biomedicines-10-02439-t003:** Characteristics of included studies according to GRADE recommendations.

Authors	Year of Publication	Country	Included Population	Intervention	Control	Effect Measure	Treatment Duration	Follow-Up
Naftali et al. [43]	2013	Israel	21 CD	Cigarette smoking: 0.5 g of dried cannabis flowers, equivalent to 11.5 mg of THC twice daily.	Cigarette smoking *w*/*o* cannabis after THC extraction.	CDAI	8 weeks	2 weeks
Naftali et al. [44]	2017	Israel	19 CD	Cannabinoid oil at a concentration of 5 mg/mL (0.3 mg/kg).	Placebo containing pure olive oil	CDAI	8 weeks	2 weeks
Matalon et al. [45]	2021	Israel	30 CD 19 UC	Cigarette smoking: 0.5 g of dried cannabis flowers, equivalent to 11.5 mg of THC.	Cigarette smoking *w*/*o* cannabis after THC extraction.	Endocannabinoid blood level change; CDAI; BM; QOL; immunohistochemistry measures.	8 weeks	None
Naftali et al. (UEG) [46]	2018	Israel	46 CD	Cannabis oil 15% cannabidiol + 4% tetrahydrocannabinol	Placebo	Clinical Remission	8 weeks	None
Irving et al. [47]	2018	United Kingdom	60 UC	Cannabidiol-rich botanical extract capsules, up to 500 mg/day	Placebo capsules	Mayo score; IBDQ score; PGAS score; stool frequency NRS; rectal bleeding NRS pain 0–10 NRS score.	10 weeks	1 week baseline + 1 week follow-up
Naftali et al. [48]	2021	Israel	32 UC	Cigarette smoking: linear increasing up to 1 g/day of dried cannabis flowers (0.25 g steps), equivalent to 23 mg of THC.	Cigarette smoking *w*/*o* cannabis after THC extraction.	Lichtiger index; QOL; Mayo endoscopic score	8 weeks	2 weeks

BM: daily bowel movements; CD: Crohn’s disease; CDAI: Crohn’s disease activity index; IBDQ: inflammatory bowel disease questionnaire; NRS: numerical rating scales; THC: tetra-hydro-cannabidiol; PGAS: physician global assessment of illness severity; QOL: quality of life questionnaire; UC: ulcerative colitis.

**Table 4 biomedicines-10-02439-t004:** RoB quality assessment of included studies. Green: low risk of bias; Yellow: some concerns in bias assessment; Red: High risk of bias.

Study ID	Domain 1	Domain 2	Domain 3	Domain 4	Domain 5	Overall
Irving 2018	Low Risk	Low Risk	Low Risk	Low Risk	Low Risk	Low Risk
Matalon 2021	Low Risk	Some Concerns	High Risk	Low Risk	Low Risk	High Risk
Naftali 2013	Low Risk	Some Concerns	Low Risk	Low Risk	Low Risk	Some Concerns
Naftali 2017	Low Risk	Some Concerns	Low Risk	Low Risk	Some Concerns	Some Concerns
Naftali 2018	Some Concerns	Some Concerns	Low Risk	Some Concerns	Some Concerns	Some Concerns
Naftali 2021	Low Risk	Low Risk	Low Risk	Low Risk	Low Risk	Low Risk

## Data Availability

All data relevant to the study are included in the article or uploaded as online Appendix A.

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
