# Peer review of "Cannabinoid Therapeutic Effects in Inflammatory Bowel Diseases: A Systematic Review and Meta-Analysis of Randomized Controlled Trials"

_biomedicines, 2022, doi:10.3390/biomedicines10102439_

Round 1

Reviewer 1 Report

1. In the background, the authors might focus on the anti-inflammatory effect of cannabinoid, instead of describe to much about pain, which is not the main point of the article.

2. Why Embase was not searched? It is recommended as the must-search database in the Cochrane handbook. I suggest the authors to give rationale for the selection of databases.

3. RR in the method section should be relative ratio instead of relative risk.

4. The administration method was different across studies, making the meta-regression unnecessary on the dosage reported in the articles.

5. As the authors stated that there is no evidence for the anti-inflammatory effect of cannabinoids, the pain reduction leading to a better quality of life may be of interest to readers. Why did the authors not assess the quality of life in the review?

Author Response

Thank you for providing a timely and accurate review of our paper. We took great care in its preparation, and we believe the content and general tone of the comments received reflect the underlying value of the work.
We thank both reviewers for their punctual remarks, which we believe will improve the article overall quality, once addressed.
Hereby, we provide a point-by-point answer to what was raised.

1) We have expanded section "1.1. Background", adding more information on the anti-inflammatory effect of cannabinoids and some more insight on the state-of-the-art. Some further references were also added when opportune.

2) Despite both Scopus and Embase being maintained by Elsevier, institutions may grant their members access to one database, but not the other, which makes impossible for us to perform a search on Embase as this was indeed our case. While it is true that Embase was not searched, we nonetheless performed the search on Scopus, since their actual difference appear to lie not in the sources used but in the search algorithm and keywords functions. A further point is to be raised, that source coverage overlaps between Embase and Scopus is unclear. Elsevier never states clearly if Scopus covers all Embase records, presumably for marketing strategy (if the answer to previous sentence was clearly yes, then a library with limited sources would subscribe to Scopus and not Embase). https://www.elsevier.com/solutions/embase-biomedical-research/learn-and-support
Since we have focused our work on RCT studies, we searched on 3 different databases, and all other sources (i.e. the other papers and previous review on the matter) substantially confirm that the existing RCT on cannabinoid supplementation are the ones actually retrieved. Even in the unlikely occurrence that a RCT was indeed performed and published on a source that is not covered by any of the 3 databases used (PubMed, Scopus, and ClinicalTrials.gov), this would put at least a suspicious light on the quality, reliability and scientific validity of said hypothetical RCT. For all these reasons, we do not believe that missing Embase has had any impact on our results.
As suggested, we amended section "2.3. Information sources and search strategies" methods to explicit that search query was performed only on accessible databases.

3) This was a typo in the methods section; corrected to "pooled Risk Ratio (RR)".

4) We thank you for this observation, since it allows us to highlight another point in our paper. While it is true that the administration routes are different, it must be noted that THC and cannabidiol pharmacokinetics are quite complex to predict: smoked cannabis absorption has a quick concentration peak but is heavily influenced intra- and inter-subject variability in smoking dynamics, which contributes to uncertainty in dose delivery. The number, duration, and spacing of puffs, hold time, and inhalation volume, greatly influences the degree of drug exposure. Oral administration is influenced by several factors, including variable absorption, degradation of drug in the stomach, and significant first-pass metabolism to active 11-OH-THC and inactive metabolites in the liver. Nonetheless, smoked peak concentration reaches quickly a low plasmatic availability (within 1h), and similar concentration were reported for standard oral intake of synthetic THC (Marinol®, 10mg THC). Since we are focusing on long-term effects of cannabinoid as anti-inflammatory adjuvant in the gastrointestinal tract, we believe that difference in administration way does not hinder the validity of our analysis, as peak concentration should only matter for brain and psychological effects. We have explicated all this, with relative references, in section "4.3. Limitations of the review methods", and built on this premise in the clinical statement added in section "4.4 implications".

5) We point out that ineffectiveness as anti-inflammatory is clear only in UC. This is not true for CD. We explicated this in the text. Quality of life (QoL) was not considered as a primary objective of the research, whose design was centred on clinical effectiveness of cannabinoid as adjuvant therapy in terms of therapeutic success. This because:
a) it is less objectivable, relying on patients' own feelings rather than lab findings or validated scales;
b) since cannabinoid usage is involved, QoL change are unreliable, especially in trials involving THC, since patients may give extra benefit to the pleasant cannabinoid-induced psychoactive effects, rather than on clinical modification of their IBD.
Nonetheless, some QoL ameliorants were found in the retrieved literature; they have been mentioned in the narrative synthesis as side results of this review in paragraph.

Reviewer 2 Report

This is a well-written and of current interest systematic review and meta-analysis on the use of cannabinoids in patients with ulcerative colitis and Crohn's disease. The methodology, data analysis, and conclusions drawn are sound and consistent with the results.

My remarks concerning the study are minor and consist of the following:

At the conclusion part of the study I would suggest to the authors and for the sake of the readers who do not have the time to go through the text in detail, to formulate more clearly what their point of view could finally be regarding whether or not cannabinoid supplementation should ultimately be administered to patients with Crohn's disease alongside conventional treatment, at what dosage, and for how long (e.g. only in the induction period or/and during the maintenance period?).

Finally, about the administration of cannabinoid supplementation as an analgesic treatment, I believe that no one dealing with the diagnosis and treatment of IBD patients uses cannabinoids as analgesics. Their use in clinical studies should only be restricted to enhancing the results of conventional treatment in causing clinical and laboratory remission. 

Author Response

Thanks for the general green light. We are happy to see the work was appreciated.

1) we have expanded section 4.4. "Implications" as requested, with our clinical opinions.

2) We fully agree with the reviewer on the point and have added a paragraph on this in section "4.4. Implications"

Round 2

Reviewer 1 Report

My questions are all answered. I have no more comments.